# LATENTS-INV: ROBUST SEMANTIC WATERMARK WITH KEY-ASSISTED RECOVERY FOR DIFFUSION MODELS

## ABSTRACT

Semantic watermarking provides imperceptible identity traceability for diffusion-generated images, enabling model copyright protection and image source verification. However, existing semantic watermarking methods based on initial latent noise render the protected image vulnerable to adversarial latent-space manipulations, such as black-box forgery via proxy models and watermark-pattern-removal attacks that exploit statistical regularities. In this paper, we propose a robust watermarking framework resilient diverse adversarial manipulation attack. Specifically, we design a fully reversible, flow-based codec with dual encoding paths, allowing plug-and-play integration into the diffusion generation process across architectures (UNet and MMDiT). The dual-output network encodes watermark information into both the carrier image and the owner's secret key, enabling recovery of removal attacked watermark via key-assisted reconstruction. To guarantee verification reliability without excessive reliance on the key while retaining the ability to detect forged watermarked images, we propose a joint-training strategy that leverages negative-sample pairs under both accuracy and fidelity constraints. Furthermore, we introduce an Euler-based enhanced solver for the effective inversion in rectified flow models, which improves the accuracy of watermark information recovered. Experimental results show that our method achieves superior robustness under various attacks while maintaining high visual quality across diverse models.

## 1 INTRODUCTION

Digital watermarking helps improve security in content creation based on strong cross-modal vision models(Xian et al., 2024; Song et al., 2024; Lei et al., 2024; Zhang et al., 2024b). Individuals with different backgrounds can train personalized models based on their own needs, resulting in various models that generate content of quality similar to human-created content(Rombach et al., 2022; Ho et al., 2020; Xu et al., 2024),. However, this open approach to model customization brings trust issues, such as model theft, copyright disputes, and the generation of fake contentZhang et al. (2024c). Watermark information can not only be used to protect model copyright, which is valuable intellectual property from large-scale training. But also to embed traceable data into the model's outputs(Bao et al., 2024; Pan et al., 2024b). This makes it possible to check the source of generated content(Min et al., 2024; Feng et al., 2024; Chen et al., 2024), helping to identify fake, false, or illegally used outputs, and ensuring that generated results are secure, reliable, and trustworthy.

For generative models of visual content, watermark information can be embedded by marking the model's outputs with a secret message(Chen et al., 2024; Pan et al., 2024b; Trias et al., 2024). Currently, the main watermarking methods fall into two categories: post-processing and in-processing approaches. Post-processing is similar to traditional image watermarking. It steganographically embeds information into the output after the model generates the data. However, this approach is easily affected by image-level detection and attacksLiu et al. (2024). In contrast, most current watermarking methods for diffusion models embed verifiable information during the generation process, at specific stages of image creation. These methods include fine-tuning parts of the model(Rezaei et al., 2024; Hu et al., 2024a) and watermarking the initial noise before the denoising process(Wen et al., 2023; Ci et al., 2024; Yang et al., 2024b).

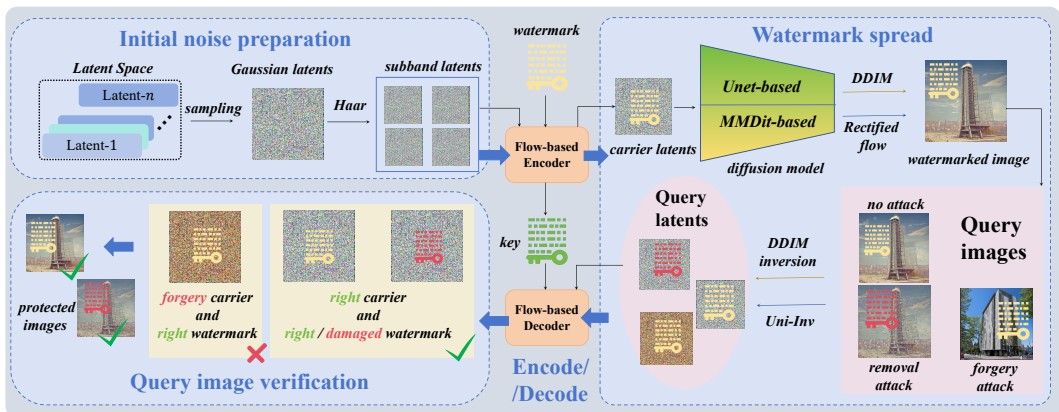

Figure 1: Framework of the proposed method system. Latents-Inv uses the sampled and transformed sub-band signals as the watermark carrier. During spreading, the watermark information may be partially removed from the denoised image or forged on an attacker's image. In the verification phase, forgery attacks do not cause confusion, and removal attacks do not prevent ownership identification.

The first method is designed for specific model architectures, it has poor generalization—watermarking methods for UNet-based models often do not transfer well to MMDiT-based models. Therefore, in this paper, we propose Latents-Inv. This method still embedds watermark information into the latent representations at the beginning of generation for various models, as shown in Figure1. This approach is theoretically convenient for achieving plug-and-play compatibility across different models. However, in practice, extracting semantic watermarks relies on accurate inversion and reversible distribution transformation. Due to differences in latent space dimensions and denoising processes across models, current watermarking approaches are not fully compatible, often leading to significant drops in robustness and generation qualityUmrajkar & Singh (2025). For UNet-based diffusion models, watermarking typically uses mature inversion methods like DDIM or DPM-Solver, which are based on ODE solversZhao et al. (2022). In contrast, more advanced models with MMDiT architecture use a Rectified Flow-based denoising process. This is first-order Euler-like method and requires iterative approach to reduce inversion errors Rout et al. (2024). To address the precision issue, our method predicts the backward-step noise from the forward-step predictions and then uses this estimate to refine the forward prediction, as detailed in 3.4.

Moreover, although several semantic watermarking methods in the noise space have been proposed, their robustness has been challenged by various watermark attacks. For generated images that require verification, traditional image transformations—such as cropping and noising—can unintentionally or deliberately damage the watermark by altering the pixel values of the image. Modern attacks, however, focus more on breaking the watermark pattern while preserving image quality, aiming to either remove or forge watermark information and thus undermine its credibilityYang et al. (2024a). Existing semantic watermarking methods show poor robustness against proxy-model-based deep perturbation attacks on single images, where watermarks can be easily erasedMüller et al. (2025). Furthermore, when an attacker has access to a large number of watermarked images sharing the same pattern, there is a risk that the watermark pattern can be statistically analyzed and stolen using a proxy model, enabling watermark removal and forgeryPan et al. (2024a).

Therefore, Latents-Inv primarily addresses the robustness issue. Its core idea is to use the owner's key to recover the damaged parts of a watermark when the carrier image is inevitably attacked, enabling fine-grained verification with sufficient accuracy. We employ a flow-based model with a bidirectional structure to reversibly embed and accurately extract watermark information. This model uses dual-channel outputs to distribute the watermark: one channel contains the initial latents with the embedded watermark, and the other contains the copyright owner's key also embedded with watermark information. To ensure that watermark verification does not overly depend on the key, we apply a joint-training strategy during training that utilizes negative sample pairs under both accuracy and fidelity constraints. Additionally, we train a coarse-grained decoder with the same architecture but different parameters to enable watermark extraction from images even without the key, ensuring

full utilization of the carrier's frequency-domain information capacity during the watermark encoding phase.

In summary, our method minimizes perturbation to the latent representations at the early generation stage, ensuring that the embedded watermark remains highly imperceptible in the final generated images. Experimental results show that current statistical attack methods cannot detect or distinguish the watermark pattern in Latents-Inv. Moreover, our method includes a robust watermark extraction system that maintains strong ownership identification capability even when the watermark in the carrier image is disturbed by attackers. Furthermore, our approach achieves lossless watermarking in diffusion models—tested on both advanced first-order Euler-based models and DDIM-based diffusion models, it achieves consistent and strong watermarking performance.

## 2 RELATED WORK

### 2.1 WATERMARK METHOD

Embedding watermark information into data as traceable metadata is highly beneficial for source verification and copyright protection. Traditional image watermarking techniques directly embed invisible watermarks into the spatial or transform domain of the target image, serving as a post-processing steganographic method for traceability(Liang et al., 2024; Müller et al., 2025; Liu et al., 2024). However, with the development and maturity of AI generative models, watermarking methods are increasingly integrated into the image generation process itselfArabi et al. (2025). This not only improves the robustness of low-perturbation watermarks but also enables attribution of generated images based on the specific watermarking method used by a model, thus protecting both content and model ownership(Saberi et al., 2023; Liang et al., 2024). In-processing approaches vary across different generative models, but the main strategies fall into two categories: one fine-tunes part of the model structure to embed watermark information during image generationFeng et al. (2024); the other embeds the watermark at the initial noise sampling stage of diffusion models, which reduces computational overhead(Yang et al., 2024b; Ci et al., 2024; Arabi et al., 2024).

### 2.2 ANTI-WATERMARK METHOD

Robustness against attacks remains a key challenge for modern semantic watermarking. Early watermark removal methods relied on simple image transformations—such as cropping, noise addition, and rotation—which physically alter the image to disrupt invisible watermarks embedded in the spatial or transform domain.Liang et al. (2024) However, with advances in neural networks and deep learning, such post-processed perturbative watermarks can be easily detected and removed by deep modelsJiang et al. (2023). Meanwhile, existing in-processing watermarking methods embed information during generation, but they often rely on fixed mathematical patterns in the initial noisy latent, leading to consistent image distributions. This makes the watermark vulnerable to statistical analysis(Zhang et al., 2024a; Yang et al., 2024a) or proxy models(Hu et al., 2024b; Müller et al., 2025) that can learn the embedding pattern, enabling watermark removal or forgery. In addition, watermarking methods based on fine-tuning parts of the generator are limited by fixed model modules and can be erased using regeneration attacks on diffusion models(Zhao et al., 2024; Liu et al., 2024). These vulnerabilities show that current watermarking approaches still lack sufficient robustness.

### 2.3 DIFFUSION MODELS

Besides security concerns, the lack of cross-architecture generalizability is another key limitation of existing watermarking methods. Most watermarking proposals are designed for older generation models—particularly diffusion models based on the UNet architecture and the DDIM sampling processMokady et al. (2023). DDIM approximates the original SDE process of DDPM—which is based on a Markov chain—as an ODE-solving processZhao et al. (2022). Although this simplification introduces some approximation error, it makes the parameters at any noising or denoising timestep explicitly known, enabling efficient inversion from the generated image back to its corresponding initial noise. And the forward Euler method for denoising follows an ODE-solving procedure, but the backward Euler method is implicit. In the inversion process, this manifests as a mismatch between the prompt embedding vectors in the forward and backward passes, necessitating an iterative

Figure 2: The core watermark encoding and decoding structure of Latents-Inv, encompassing pre-processing, encode-decode, and post-processing stages. Pre-processing primarily employs mathematical techniques for channel expansion, thereby increasing the embeddable regions for watermark information. The flow-based encoder manages the forward process, sharing model weights with the corresponding flow-based decoder during the reverse process, while maintaining an identical architecture to that of the independent decoder. Post-processing focuses on feature fusion, ensuring dimensional consistency between inputs and outputs, as well as preserving the distribution characteristics of the latents.

solver to achieve consistency.As a result, watermarking methods that rely on fine-tuning model components fail to achieve consistent embedding and protection performance when applied to newer models based on the MMDiT architecture and rectifield-flow methodLiu et al. (2022). Similarly, approaches that embed watermarks in the initial noise suffer significant performance degradation due to architectural and procedural differences in the generation pipelineUmrajkar & Singh (2025). Specifically, models using the rectifield-flow method no longer employ only ODE-based sampling like DDIM for noising or denoising. This necessitates a new inversion method to reliably map generated images back to their corresponding initial noise(Wang et al., 2024; Deng et al., 2024; Rout et al., 2024; Jiao et al., 2025), ensuring stable watermark embedding and extraction within the initial noise space.

## 3 METHOD

### 3.1 OVERVIEW

As shown in Figure2, during encoding, the core of our method is using a flow-based model to embed watermark information into two outputs: the carrier latents and a owner's key. After generation and inversion, Latents-Inv should verify the query latents. In the extraction phase, fine-grained and high-precision watermark recovery is achieved with the aid of the key, while a key-free coarse-grained decoder assesses the level of watermark corruption in the carrier, helping to verify the legitimacy of the fine-grained extraction result and preventing false acceptance of forged or unauthorized content. Next, we present a detailed description of the codec operational mechanism and training strategy, along with the Uni-inv inversion method tailored for MMDiT-based diffusion models.

### 3.2 LATENTS PROCESSING AND FLOW-BASED CODEC

The flow-based model is inherently reversible: for every forward encoding function $f_\theta$, there exists a corresponding backward function $f_\theta^{-1}$. This one-to-one invertibility naturally aligns with the watermarking pipeline, where embedding and extraction are symmetric processes. However, before any forward or backward pass through the flow model, as illustrated in Figure 2, the input must go through pre-processing and post-processing.

During pre-processing, Latents-Inv will transform the latent noise into four frequency domains through two-dimensional HARR transformation. Embedding the watermark information multiple times into the HARR-transformed information can better utilize the channel capacity than embedding it once in the spatial domain. And watermark can be more robust without affecting the quality(Wen et al., 2023; Kassis & Hengartner, 2025).

In the embedding stage, Latents-Inv accepts the watermark information $\mathbf{m_0}$ and the carrier information $\mathbf{x_0}$ . And in each invertible block, the model performs additive affine transformations to gradually blend the watermark with the carrier image. Finally, it outputs the watermarked carrier $\mathbf{x_n}$ and the owner's key $\mathbf{m_n}$.Here is the invertible block:

$$x_{i+1} = x_i + U_i(m_i) \tag{1}$$

$$m_{i+1} = m_i \otimes \exp\left(D_i^1(x_{i+1})\right) + D_i^2(x_{i+1}) \tag{2}$$

where $U_i$ denotes the update network, $D_i^1, D_i^2$ are diffusion sub-networks, and $\otimes$ indicates element-wise multiplication. In the extraction stage, we input the key $\mathbf{r_n}$ and the watermark carrier $\mathbf{y_n}$ through backward input, and extract the watermark information by performing the same parameter convolution and reverse coupling of the carrier and key feature information precisely. The corresponding backward propagation of the extraction process is formulated as:

$$m_i' = \left(m_{i+1} - D_i^2\left(x_{i+1}'\right)\right) \otimes \exp\left(-D_i^1\left(x_{i+1}'\right)\right) \tag{3}$$

$$x_i' = x_{i+1}' - U_i(m_i') \tag{4}$$

At the same time, we also use a trained coarse-grained watermark decoder with the same structure, inputting an empty matrix $\mathbf{z_n}$ and the watermark carrier $\mathbf{y_n}$, and directly extracting the watermark information from the carrier. Coarse-grained occurs before fine-grained extraction, it can ensure the validity of the carrier information and ensure that fine-grained extraction does not overly rely on the redundant residual watermark information in the key.

Post-processing ensures consistent data dimensions and helps preserve output image quality. In the whole process, the key is processed by a fully connected (fc) layer to match the same dimension as the original watermark. For carrier latents, Latents-Inv concatenates the clean and watermarked frequency-domain signals across the four bands along the channel dimension, followed by a 1×1 convolution.This ensures that the initial noise fed into the diffusion model contains the watermark and better matches the desired noise distribution.

### 3.3 WATERMARK ROBUSTNESS PROTECTION

To address the issue of robustness, our method hides the watermark information through an invertible latent watermark encoder-decoder in two parts: the released carrier image and the private key owned by the image owner. Since the watermark information associated with the private key remains intact, our method is inherently robust against attacks that attempt to corrupt the watermark in the carrier. To ensure reliable verification, however, it is essential to have sufficient discriminative capability over the distribution of latents paired with the key.

First, we evaluate watermark embedding and extraction under noise-free conditions, which serves as the fundamental criterion for assessing embedding and decoding accuracy. Unlike prior semantic watermarking methods, Latents-Inv preserves the distribution of the discretized latent noise as much as possible during modification, thereby maintaining content consistency of the generated images before and after watermark embedding. Our joint training strategy for image generation quality and accuracy is as follows:

$$\mathcal{L}_\theta = \mathbb{E}_{m_0, \boldsymbol{x}_0}\left[\|\boldsymbol{x}_0 - \pi_{\boldsymbol{x}} \circ f_\theta(m_0, \boldsymbol{x}_0)\|^2 + \|m_0 - \pi_m \circ f_\theta^{-1} \circ f_\theta(m_0, \boldsymbol{x}_0)\|^2\right] \tag{5}$$

where $\pi_{\boldsymbol{x}}(\boldsymbol{x}, m) = \boldsymbol{x}$ and $\pi_m(\boldsymbol{x}, m) = m$ are the natural projections. We avoid using KL divergence as the distribution loss because it may not sufficiently constrain the $L_2$ distance between the initial latents before and after watermark embedding, potentially leading to perceptible distortions in the generated image.

To enhance robustness against watermark destruction attacks, we leverage the multi-band capacity of the initial latent noise and employ a fully reversible, dual-end flow-based architecture. To prevent

the model from over-relying on the key for watermark encoding—thus underutilizing the latent space—we introduce a coarse-grained structural decoder. It takes a zero matrix and the carrier noise as input and reconstructs the initial watermark, promoting watermark embedding in the latent channel while preserving the integrity of the carrier image. The corresponding loss is:

$$\mathcal{L}_{\theta'} = \mathbb{E}_{m_0, \boldsymbol{x}_0, z_0} \big\| m_0 - \pi_m \circ f_{\theta'}^{-1} \circ \big( f_\theta(m_0, \boldsymbol{x}_0), z_0 \big) \big\|^2 \tag{6}$$

For watermark forgery through the published carrier image, we introduce negative samples pairs through regularization penalties, so that the trained flow-based model pays more attention to the noisy latents with key-paired watermark information during decoding. We employ the BCE loss as a metric to measure the wrong classification rate on negative sample pairs, and introduce a penalty term to prevent overfitting:

$$\mathcal{L}_{bce} = BCE(m_0, \pi_m \circ f_\theta(m_0, \boldsymbol{x}_0)) \tag{7}$$

$$P_{\text{neg}} = \max\big(0, \ 0.6930 - (\mathcal{L}_{\text{bce}} - \mathcal{L}_{\theta'})\big) \tag{8}$$

The penalty threshold is set to 0.6930, which corresponds to the BCE loss of a random guess on a balanced binary watermark sequence. At the same time, negative sample pairs prevent the decoder from relying too much on the key, ensuring the latent noise is also used for watermark extraction. This avoids false detection when arbitrary noise is paired with a key.

In brief, to balance watermark embedding between the carrier and the key, we combine coarse- and fine-grained decoding with negative-sample training. The overall training loss with the corresponding weight coefficients $\lambda_1$, $\lambda_2$ and $\lambda_3$ is:

$$\mathcal{L}_{\text{total}} = \lambda_1 \mathcal{L}_\theta + \lambda_2 \mathcal{L}_{\theta'} + \lambda_3 P_{\text{neg}}. \tag{9}$$

### 3.4 INVERSION METHOD IN RECTIFIELD-FLOW DIFFUSION MODEL

Diffusion-based generative models aim to map initial noise samples drawn from a Gaussian distribution to realistic data distributions. The inversion process seeks to reconstruct the reverse diffusion trajectory in order to recover the initial latent noise corresponding to a given generated image—a critical step for accurately extracting watermarks embedded via latent noise modulation. However, recent models such as SD3 and FLUX adopt Rectified Flow formulations rather than DDIM-style ODE solvers:

$$Z_{t_{i-1}} = Z_{t_i} + (t_{i-1} - t_i) v_\theta(Z_{t_i}, t_i) \tag{10}$$

where $Z_{t_{i-1}}$ denotes the latents at timestep $t_i$, and $Z_{t_i}$ is the latents from the previous step. The term $t_{i-1} - t_i$ represents the time step size, and $v_\theta(Z_{t_i}, t_i)$ is the velocity network that predicts the flow direction at $t_i$. This explicit forward Euler update simulates the continuous transformation from data to Gaussian noise in Rectified Flow diffusion models. So, the inversion process may clear:

$$Z_{t_i} = Z_{t_{i-1}} - (t_{i-1} - t_i) v_\theta(Z_{t_i}, t_i). \tag{11}$$

The variable $v_\theta(Z_{t_i}, t_i)$ is unknown in the backward Euler method. Consequently, the inversion method cannot be straightforwardly derived from the forward process, due to the non-ODE formulation and implicit backward dynamics of modern flow-based models. Therefore, in the experiment, we adopted the precise inversion method of Uni-InvJiao et al. (2025), which reverses the previous value of hidden Euler approach back to the next value. Given the velocity function $v_0$, initial image $Z_0 \sim \pi_0$, and time steps $t = \{t_0, \ldots, t_N\}$ where $t_0 = 0$ and $t_N = 1$, the Uni-Inv (Euler) algorithm updates are defined as follows:

Initial conditions:

$$\widehat{v}_0 = v_0(Z_0, t_0) \tag{12}$$

$$\widehat{Z}_{t_0} = Z_0 \tag{13}$$

For $i = 1$ to $N$:

$$\overline{Z}_{t_i} = \widehat{Z}_{t_{i-1}} - (t_{i-1} - t_i)\widehat{v}_{i-1} \tag{14}$$

$$\widehat{v}_i = v_\theta(\widehat{Z}_{t_i}, t_i) \tag{15}$$

$$\widehat{Z}_{t_i} = \widehat{Z}_{t_{i-1}} - (t_{i-1} - t_i)\widehat{v}_i \tag{16}$$

The final output we want is $\widehat{Z}_1$. Uni-Inv is fundamentally an iterative generation method designed for the backward Euler ODE solver, also known as the predictor-corrector method—a classical and effective approach in numerical analysis for solving differential equations. The theoretical error is $\mathcal{O}(\Delta t_i^3)$, where $\Delta t_i = t_i - t_{i-1}$, and the proof is shown in the appendix. It transitions to the high-noise step first, estimates the velocity by simulating a denoising process. Then it returns to the original low-noise step and performs inversion using the latest "denoising-like" velocity. Intuitively, because the initial and end noise distributions are similar, the parallel noise distribution errors on the same time step with the same length will be smaller.

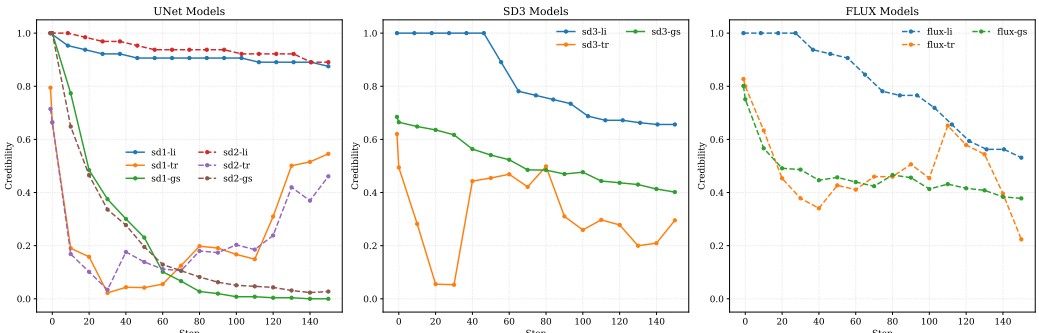

Figure 3: Robustness of semantic watermarking methods under surrogate-model-based removal attacks across different diffusion architectures.

## 4 EXPERIMENTS

### 4.1 EXPERIMENTAL SETUP

**Datasets.** In our experiments, we use a combination of real and synthetic images. We randomly select 5,000 real images from COCO2017 as clean, unwatermarked samples to serve as the reference pattern in detection. We generate 19,000 clean images and 1,000 watermarked images with SD3, where 10% of the watermarked set is used as training data. All images are generated from prompts in the Stable-Diffusion-Prompt dataset and are also used to evaluate detection performance.

**Diffusion Models.** We evaluate across four diffusion models: SD1.5 as a representative UNet-based architecture, and SD3.medium and FLUX1.dev for testing under MMDiT and rectified flow (RF) generation frameworks. For black-box forgery and removal attacksMüller et al. (2025), we adopt SD2.1-baseRombach et al. (2022) as the surrogate model.

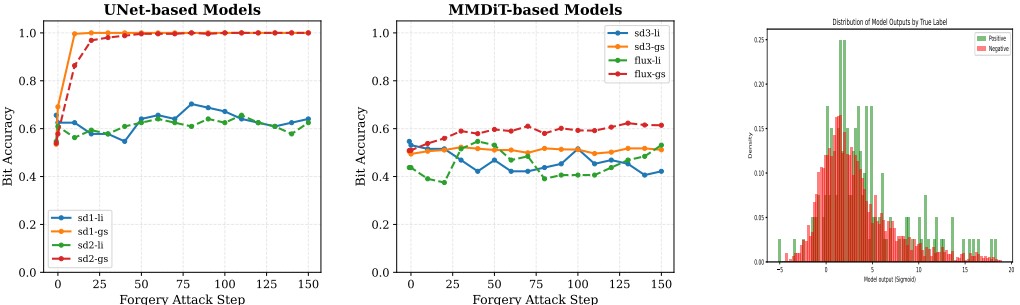

Figure 4: Robustness of semantic watermarking methods under surrogate-model-based forgery attacks across different diffusion architectures.

Figure 5: Distribution of images after statistical analysis detection.

**Watermarking and Adversarial Methods.** We compare with semantic watermarking methods that embed information in the initial latent noise, focusing on those transferable from UNet to MMDiT architectures. We select Tring-Ring (TR)Wen et al. (2023) and Gaussian Shaping (GS)Yang et al. (2024b) as baselines. To assess robustness, we consider two attack paradigms: (i) single-image

latent inversion attacks that perturb initial latentsMüller et al. (2025), and (ii) statistical detection methods that analyze patterns across large collections of watermarked imagesYang et al. (2024a); Pan et al. (2024a).The former can perform threatening watermark removal and forgery attacks and the latter can effectively distinguish the distribution of watermarked images from clean images.

**Evaluation Metrics.** We use **bit accuracy** as the primary metric for GS and Latents-Inv watermark extraction robustness. We measure TR watermark extraction robustness with the p-value, which gives the probability that the observed watermark would appear by random chance—smaller values indicate stronger evidence of the true watermark. And for monotonicity consistency, we use 1-p-value for comparison.

Table 1: Comparison of Different Methods

| Method | Model | Clean | Crop | Noise | Bright | Removal | Forgery | Avg ↑ |
|---|---|---|---|---|---|---|---|---|
| Tree-Ring | SD1.5 | 0.963 | 0.067 | 0.522 | 0.631 | 0.546 | 0.000 | 0.455 |
| | SD2.1 | 0.978 | 0.079 | 0.544 | 0.664 | 0.461 | 0.000 | 0.454 |
| | SD3.0 | 0.694 | 0.032 | 0.312 | 0.353 | 0.295 | 0.005 | 0.282 |
| | FLUX.1 | 0.722 | 0.041 | 0.383 | 0.419 | 0.223 | 0.017 | 0.301 |
| Gaussian-Shading | SD1.5 | 1.000 | 0.847 | 0.693 | 0.864 | 0.003 | 0.000 | 0.568 |
| | SD2.1 | 1.000 | 0.848 | 0.715 | 0.825 | 0.027 | 0.000 | 0.569 |
| | SD3.0 | 0.787 | 0.648 | 0.576 | 0.626 | 0.402 | 0.489 | 0.588 |
| | FLUX.1 | 0.804 | 0.683 | 0.608 | 0.705 | 0.322 | 0.386 | 0.585 |
| Latents-Inv (Ours) | SD1.5 | 1.000 | 0.821 | 0.724 | 0.869 | 0.906 | 0.359 | 0.777 |
| | SD2.1 | 1.000 | 0.831 | 0.745 | 0.872 | 0.856 | 0.375 | 0.780 |
| | SD3.0 | 1.000 | 0.812 | 0.728 | 0.822 | 0.656 | 0.593 | 0.769 |
| | FLUX.1 | 0.984 | 0.804 | 0.687 | 0.794 | 0.781 | 0.532 | 0.764 |

## 4.2 EXPERIMENTAL RESULTS

**Watermark Robustness Performance of Latents-Inv:**Table 1 summarizes robustness evaluation results under both traditional image distortions and black-box watermark removal attacks using the SD2.1-based surrogate model. Under conventional attacks, semantic watermarking methods based on initial latent noise generally show reliable performance. Our method achieves competitive or superior bit accuracy across most cases. Notably, while Latents-Inv performs slightly worse than GS under cropping on UNet-based models, our approach maintains consistent robustness. Even when the latent channel dimension increases from 4 to 16, our method preserves the highest detection rate, demonstrating minimal interference during model transfer and achieving over 98% accuracy on both SD3 and FLUX1.dev.

Under black-box attacks via latent space manipulation, our method shows strong resilience. As shown in Figure 3, with increasing optimization steps in the surrogate model, all semantic watermarks suffer some degradation. GS degrades rapidly on U-Net-based models, whereas our method maintains over 85% accuracy, showing stable robustness. On MMDiT models, our watermark remains robust in early stages and degrades gracefully, still outperforming baselines. Interestingly, TR, which embeds watermark in a ring-like structure similar to physical patterns, exhibits oscillating behavior under iterative perturbation—consistent with the nature of the optimization-based attack. A detailed analysis is provided in the Appendix.Figure 4 shows results under black-box forgery attacks using a surrogate model, where the ring-based non-bit-embedding watermark (TR) is excluded for clarity. On UNet-based models, our method significantly outperforms GS. On MMDiT models, our approach also achieves better robustness, though the performance gap is smaller. Notably, due to architectural differences and limitations of the surrogate model, such forgery attacks struggle to effectively manipulate semantic watermarks across different model architectures—indicating limited cross-structure transferability of current latent space manipulation techniques.

**Watermark Visual Performance of Latents-Inv:**Figure 6 shows images generated by different diffusion models using various initial latent codes under the same prompt. SD3 and FLUX generate images at resolutions of $512 \times 512$ and $1024 \times 1024$, respectively, requiring latents of different dimensions—enabling us to evaluate watermarking under varying latent spaces. Visually, our method

produces results most consistent with the clean (unwatermarked) images, thanks to a distribution-preserving loss used during embedding. Figure 5 also presents results from a black-box statistical detection test on Latents-Inv. Negative indicates watermarked images, while Positive represents clean images. The detector fails to reliably distinguish between watermarked and unwatermarked images, indicating that our embedded latents closely match the distribution of clean ones. This explains the high visual fidelity of the generated outputs.In the last row of Figure 6, we show the pixel-wise difference between images generated from watermarked and clean latents. The residual signal is widespread across the entire image space, demonstrating that Latents-Inv embeds rich, spatially distributed watermark information. This full-spatial presence of watermark traces contributes significantly to its robustness—ensuring detectable signals remain even after partial corruption or attacks.

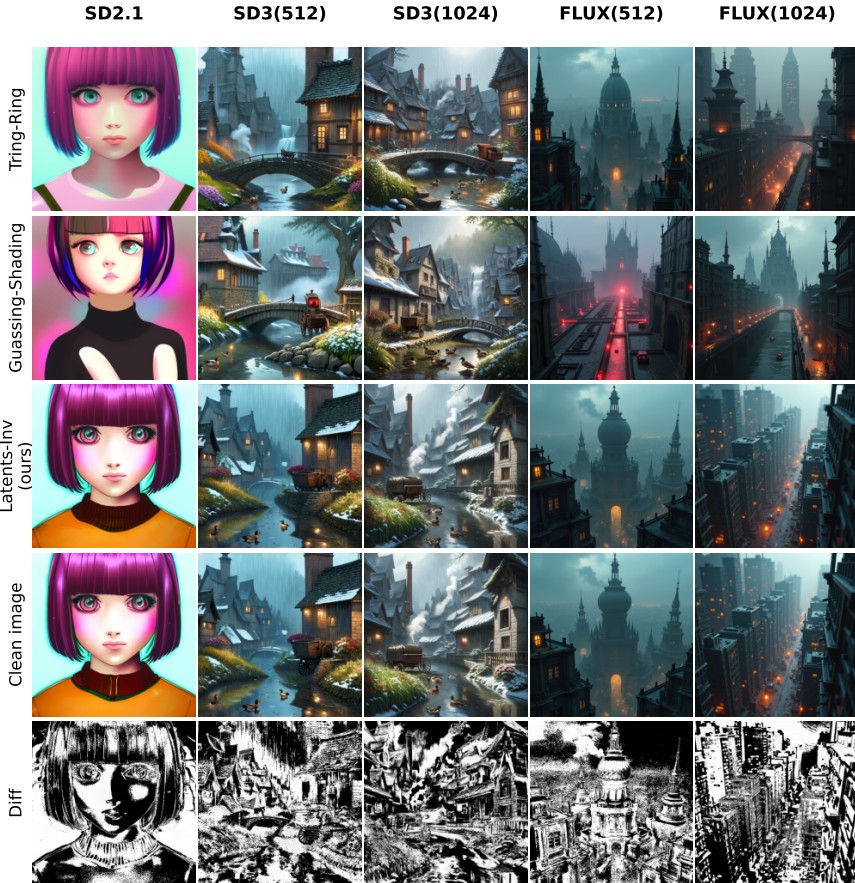

Figure 6: Comparison of semantic watermark images generated by different models at various resolutions. The first three rows represent watermarked images, the fourth row shows clean (unwatermarked) images for reference, and the last row highlights watermark artifacts produced by the Latents-Inv method.

## 5 CONCLUSION

In this work, we propose Latents-Inv, a semantic watermarking method for diffusion models that embeds invertible watermarks into the initial latent space. By employing a joint-training strategy that leverages negative-sample pairs under both accuracy and fidelity constraint for a flow-based codec. Our method achieves high visual fidelity and strong robustness across diverse architectures, including UNet and MMDiT. Extensive experiments show that Latents-Inv outperforms existing methods under both traditional distortions and black-box removal/forgery attacks, demonstrating superior transferability and resilience.

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
