# OpenReview forum: "Latents-Inv:Robust Semantic Watermark with Key-Assisted Recovery for diffusion models"
_ICLR.cc/2026/Conference — ICLR 2026 Conference Withdrawn Submission_

### Official Review · Reviewer_d9er · 2025-10-21

**Soundness:** 2
**Presentation:** 1
**Contribution:** 2
**Rating:** 2
**Confidence:** 5

**Summary:**

This work proposes a new watermarking method for diffusion models. The key idea is to train a codec/decoder to transform the input "sampled latents + watermark" into output "new latents + key", and then diffusion on the new latents to generate images. The authors show that their method is robust to various common image distortions compared to Tree-Ring and Gaussian-Shading.

**Strengths:**

+ The new method addresses the robustness issues mainly by retaining a key that goes through the codec together with the latents. So this key essentially captures watermark information together with the transformed latents. If an image is distorted, leading to watermark damage, then the above key retained by the owner could help to restore watermark information, hence improving robustness.

+ Compared to the recent training-free works Tree-Ring and Gaussian-Shading, this work Latents-Inv indeed improves robustness on various image transformations.

**Weaknesses:**

- The paper is written and presented in a poor way, making it challenging for readers to follow and appreciated their contribution (see my questions and suggestions below).

- The design of the codec/decoder network architecture lacks explanation.

- The usage of retained key seems to be a double-edged sword. It leaves a complicate question of how to store and manage such keys, and how to match a retained key with a query image during detection.

- The experimental comparison is not sufficient.

- The fidelity of generated images should be evaluated in more formal and quantifiable metrics.

**Questions:**

- The codec architecture is unclear. In equation (1) and (2), what's update network? What's diffusion sub-networks? Could you provide intuition and explanation on the block design? How do you finally achieve such formulas?

- In equation (3) and (4), these symbols do not align with Figure 2.

- How do you manage the output private keys? Given a query image, how to you match it with a key? Do you enumerate all keys in a database?

- Give more intuition on Equation (5).

- Explain equation (7). It is hard to connect this with the text about negative sample pairs.

- Expand the computation of the magic number "0.6930" in equation 8.

- This work, as a training-based watermark, should also compare with Stable Signature, which is also a training-based watermark. It cannot just compare with training-free methods Tree-Ring and Gaussian-Shading.

- The experiment in Figure 6 is good but informal.  You should give a quantifiable metric across a dataset for fidelity evaluation (such as FID), instead of just showing results for several examples.

---

### Official Review · Reviewer_TCpC · 2025-10-21

**Soundness:** 2
**Presentation:** 1
**Contribution:** 2
**Rating:** 2
**Confidence:** 4

**Summary:**

The paper proposes Latents-Inv, a semantic watermarking framework for diffusion models. It uses a fully reversible, flow-based codec with dual outputs to embed watermark information jointly into (i) the carrier’s latents and (ii) an owner’s private key, enabling key-assisted recovery when a watermark is damaged. To support newer rectified-flow models (e.g., SD3/FLUX), the paper adopts an Euler-style “Uni-Inv” inversion procedure for watermark extraction.

**Strengths:**

Timely problem & cross-architecture aspiration. The paper targets robustness to modern removal/forgery attacks and explicitly aims to work across UNet and MMDiT/rectified-flow architectures, a relevant and important goal for the community.

**Weaknesses:**

1. Incomplete submission (Appendices missing). The manuscript cites appendices for essential proofs/analyses, but no appendix is present in the provided materials/supplement, making key claims unverifiable.

2. No objective visual quality metrics. Evaluation focuses on watermark bit-accuracy without standard image visual quality metrics (e.g., CLIP-Score, FID), so the trade-off between robustness and perceptual quality is unclear.

3. Ablation studies are absent. There is no ablation isolating contributions (e.g., dual-output codec vs. single-channel, key-assisted recovery, coarse- vs. fine-grained decoder), leaving the source of gains ambiguous. (No “ablation” section appears in the provided manuscript.)

**Questions:**

See Weaknesses.

---

### Official Review · Reviewer_4QJ1 · 2025-10-30

**Soundness:** 2
**Presentation:** 1
**Contribution:** 2
**Rating:** 2
**Confidence:** 3

**Summary:**

The paper proposes Latents-Inv, a semantic watermarking framework for diffusion models that embeds the watermark in the initial latent space via a reversible flow-based codec with a dual-output design: one branch goes into the carrier latents that generate the image, and the other is stored as a private key. When the carrier watermark is degraded by attacks, the key enables key-assisted recovery to improve robustness. Training uses a joint objective with negative sample pairs to limit over-reliance on the key while enabling forgery rejection under accuracy and fidelity constraints. The paper also adopts Uni-Inv (Euler) for Rectified-Flow models to improve inversion accuracy, enabling extraction across UNet and MMDiT backbones. Experiments on SD1.5/2.1/3 and FLUX under multiple attacks suggest improved robustness over existing methods with small visual differences from the originals.

**Strengths:**

1. The combination of a reversible flow codec, dual channels (carrier/key), and coarse/fine decoders supports recovery while suppressing overfitting to the key; the overall design is coherent.

2. The paper discusses inversion differences between UNet/DDIM and MMDiT/Rectified-Flow and uses Uni-Inv (Euler) to stabilize RF inversion, aiding generality.

3. Reported results outperform prior methods, with minor visual differences between watermarked and original images.

**Weaknesses:**

1. Writing is difficult to follow. The logical flow is unclear, transitions are abrupt, and key concepts (e.g., invertible block, update network, diffusion sub-networks, flow-based codec) are insufficiently defined. Several statements are vague (e.g., “Pre-processing primarily employs mathematical techniques for channel expansion”) without a concrete explanation.

2. Figures 1 and 2 are not self-explanatory and lack clear guidance, making it hard to understand the method.

3. The dual-output design implies better decoding with the key. Although a coarse key-free decoder is provided, the paper does not report its performance under attacks. This raises the concern that without the key and under attack, detection may fail. The paper should quantify key dependence across scenarios.

4. Insufficient analysis of compute/latency: extra memory and compute for the flow codec, the impact of inversion steps on throughput, and compatibility with production samplers/schedulers are not adequately evaluated. The work focuses on accuracy without a cost–benefit profile.

5. Image quality is shown only via visuals; quantitative metrics (e.g. FID) are missing.

**Questions:**

1. What are the degradation curves of the fine decoder when the key is partially corrupted, fully missing, or tampered with? How robust is the coarse-grained decoder (no-key) against various attacks?

2. What are the additional memory and compute overheads for encoding/decoding and for training?

3. What are the FID (and related) comparisons between original and watermarked images?

---

### Official Review · Reviewer_ASUC · 2025-11-01

**Soundness:** 3
**Presentation:** 3
**Contribution:** 3
**Rating:** 2
**Confidence:** 3

**Summary:**

The authors propose a robust and invertible watermarking framework that acts directly on the diffusion process. The authors introduce a flow-based model which achieves consistency between the latent and key that ultimately helps to protect against forgery attacks.

**Strengths:**

- Integrates into a wide class of models (Uiet and MMDiT)

- Empirical results seem kind of promising

- I think flow-based models do make sense for this task and the entangling of the key + latent creates mutual consistency that allows for forgery and damaged watermark detection

**Weaknesses:**

- Baselines are very limited although Gaussian-Shading  & Tree-Ring are good to include. I would be interested in performance against Seal, Aqualora, and etc.

- Dataset is kind of limited (more kinds of images would be better)

- Formatting and typo issues make it difficult to read. Please be precise about how you cite work in your paper. Difference between \citet and \citep (feels rushed).

- Attacks are way to simple. Please include regeneration, rotation, and etc (as well as more difficult combination attacks). WAVES provides a good benchmark for this (not all are needed)

- Although the method is new, besides the meshing of the key + latent with a flow network it doesn't feel all the novel. And that too has been done in the past in a different way.

- No image quality metrics presented

**Questions:**

- I don't think that the architectural backbone is very relevant as many modern watermarking methods don't care about the architecture bur rather the distribution of the latent space and DDIM sampling. Maybe I am misunderstanding this claim.

- I wonder if training is very expensive in practice because of the need for feedback from the diffusion models.

---

### Note · Authors · 2025-12-03

I have read and agree with the venue's withdrawal policy on behalf of myself and my co-authors.